# Proximal Median Nerve Compression in the Differential Diagnosis of Carpal Tunnel Syndrome

**DOI:** 10.3390/jcm11143988

**Published:** 2022-07-09

**Authors:** Pekka Löppönen, Sina Hulkkonen, Jorma Ryhänen

**Affiliations:** 1Department of Orthopedics and Traumatology, Seinäjoki Central Hospital, FI-60220 Seinäjoki, Finland; 2Department of Hand Surgery, Helsinki University Hospital, University of Helsinki, FI-00029 Helsinki, Finland; sina.hulkkonen@hus.fi (S.H.); jorma.ryhanen@hus.fi (J.R.)

**Keywords:** carpal tunnel syndrome, median neuropathy, median nerve entrapment, neuralgic amyotrophy, pronator syndrome

## Abstract

Carpal tunnel syndrome (CTS) is the most common median nerve compression neuropathy. Its symptoms and clinical presentation are well known. However, symptoms at median nerve distribution can also be caused by a proximal problem. Pronator syndrome (PS) and anterior interosseous nerve syndrome (AINS) with their typical characteristics have been thought to explain proximal median nerve problems. Still, the literature on proximal median nerve compressions (PMNCs) is conflicting, making this classic split too simple. This review clarifies that PMNCs should be understood as a spectrum of mild to severe nerve lesions along a branching median nerve, thus causing variable symptoms. Clear objective findings are not always present, and therefore, diagnosis should be based on a more thorough understanding of anatomy and clinical testing. Treatment should be planned according to each patient’s individual situation. To emphasize the complexity of causes and symptoms, PMNC should be named proximal median nerve syndrome.

## 1. Introduction

Carpal tunnel syndrome (CTS) is the most common median nerve compression neuropathy. As its clinical symptoms and presentation are well known, the correct diagnosis and treatment are evident. However, symptoms at median nerve distribution can also be caused by nerve compression proximal to the carpal tunnel. This should be remembered when symptoms are atypical to CTS or persist after carpal tunnel release (CTR).

Proximal median nerve compression (PMNC) is more uncommon than CTS and probably underdiagnosed. Diagnosis can be difficult due to overlapping symptoms with CTS; moreover, both can coexist in the same patient [1,2,3,4,5,6]. In addition, multiple anatomical features can cause entrapment of the median nerve, presenting various symptoms. Therefore, successful treatment of PMNC requires a thorough understanding of the median nerve’s anatomical aspects, comprehension of its pathology and recovery, and experience interpreting the varying clinical presentations.

Only limited good-quality clinical research exists on PMNC. Most studies are small hospital-based series of surgically treated patients. No comparative trials have been published on treatment of PMNC. The largest problem with PMNC is the difficulty in differentiating whether or not it is a compressive neuropathy.

A literature search was performed in November 2021 in PubMed/Medline. Terms “Pronator syndrome”, “Lacertus syndrome”, “Supracondylar process syndrome”, “Proximal median nerve compression”, and “Anterior interosseous nerve syndrome” were used. Abstracts of published articles were read, and full-text versions were reviewed if they seemed relevant to the subject. All papers reporting data such as symptoms, clinical diagnosis, treatment, or outcome after treatment were selected. Articles focusing on only diagnostic studies and presenting no clinical features were not included. Only articles published in the English language were included. Only articles accessible at our facility were included.

The aim of this review is to summarize the existing literature on PMNC, examine the variations of this condition, identify related symptoms and signs, and help physicians in diagnosing and treating PMNC. This paper does not discuss median neuropathies more proximal in the shoulder such as cervical radiculopathy or thoracic outlet syndrome.

## 2. Anatomy and Sites of Median Nerve Compression

The literature is filled with different terminology representing a simplified understanding of the condition. The same diagnoses with different criteria have been accepted and used by many clinicians and authors. However, there are reasons why this traditional division should be reconsidered. Here, we guide the readers across the course of the median nerve and the compressive points and etiologies of PMNC discussed in the literature.

Median nerve fibers can be traced back to cervical roots of C5-C8 and thoracic root Th1. C5-C7 form the lateral cord, and C8 and Th1 form a smaller medial cord. Lateral and medial cords form the median nerve. The nerve then travels across the axilla and the medial side of the arm with the brachial artery between the biceps brachii and brachialis muscles.

### 2.1. Supracondylar Process, Ligament of Struthers, and Supracondylar Process Syndrome

Figure 1 shows the anatomy of the elbow. A supracondylar process or a bony spur about 3–6 cm proximal to the medial epicondyle is present in 1–2% of individuals. The ligament of Struthers is a fibrous arch between this process and the medial epicondyle [7], and the median nerve and brachial artery travel underneath it if it exists [8,9,10]. The median nerve can become compressed under this bony process or the ligament [9,11,12]. This neuropathy is known as supracondylar process syndrome [8,9,12,13]. The ligament of Struthers has been described to occur and cause compression of the median nerve even without the presence of a clinical supracondylar process [13,14,15].

### 2.2. Lacertus Fibrosus and Lacertus Syndrome

Bicipital aponeurosis or lacertus fibrosus originates from the biceps brachii muscle and joins the fascia of the pronator-flexor mass (Figure 2). The median nerve and brachial artery pass underneath and are prone to compression in forearm pronosupination [2,6,11,14,15,16,17,18,19,20]. Persisting median artery has also been described to cause median nerve compression [6].

Lacertus syndrome (LS) and its symptoms in baseball pitchers were first described by Bennett in 1959 [21]. It has more recently been popularized by Hagert and Lalonde [22]. Symptoms of LS are described as a loss of key and pinch strength, loss of fine motor skills, and sense of clumsiness. Hagert states that patients with LS have (1) weakness in median nerve innervated muscles distal to lacertus fibrosus, (2) pain when compressing the nerve at the level of lacertus fibrosus, and (3) a positive scratch collapse test. These patients rarely have paresthesia in the median nerve innervated hand [23]. More simple anatomical compression tends to lead to more accessible operative treatment.

### 2.3. Pronator Teres Muscle and Pronator Syndrome

Branches of the median nerve pass the pronator teres (PT), flexor digitorum superficialis (FDS), flexor carpi radialis (FCR), and palmaris longus (PL) before passing through the PT muscle (Figure 1) [24]. In most people, the median nerve passes between the humeral (superficial) and ulnar (deep) head of PT, the prevailing location of the PMNC; in some people the latter is missing, and the nerve passes only under the humeral head of the muscle. Rare variations of the nerve passing behind the ulnar head or through the humeral head of PT have been described [25]. Thickened tendinous bands, fibrous arches, and intramuscular bands can arise from the muscles [2,6,11,15,16,17,18,20,24,25,26,27]. Hypertrophy of the PT muscle might impact the compression [17,25].

Median nerve compression by the PT muscle was first reported by Seyffarth in 1951 [28]. He suggested using the term pronator syndrome (PS), which remains the most common term to describe PMNCs. By PS, most authors mean various combinations of symptoms that usually include proximal volar forearm pain at the region of the PT muscle, with varied median neuropathy such as weakness and sensory changes. Symptoms are typically provoked by strenuous repetitive activity such as forearm pronosupination.

### 2.4. Flexor Digitorum Superficialis Arch and Superficialis/Pronator Syndrome

After PT, the nerve courses between the muscles flexor digitorum profundus (FDP) and FDS. In up to 75% of forearms, the FDS muscle has a fibrous leading edge that can be tight while covering the median nerve and anterior interosseous nerve (AIN) (Figure 3) [2,11,15,16,17,18,20,24,29]. Even elbow extension alone can cause median nerve and AIN compression at the FDS arch in certain forearms [29]. To further distinguish and specify the compressive point over the median nerve and the very close relation between FDS and PT, the term superficialis/pronator syndrome was recently proposed by Tang [30].

### 2.5. Anterior Interosseus Nerve and AIN Syndrome

AIN branches from the main median nerve at about the level of PT and innervates FDP2, flexor pollicis longus (FPL), and pronator quadratus (PQ). FDP3 is usually innervated by AIN but can be partially innervated by the ulnar nerve [31]. There are variations on the origin of AIN, however. When branching from the radial side of the median nerve, it is more susceptible to compression of musculoaponeurotic arches than when it originates from the posterior side [16]. A cadaveric study showed that AIN can easily be intraneurally separated from the main median nerve well above medial epicondyle level, even though the actual branching happens much more distally. The branch to the FCR arose from the AIN when dissected proximally [32]. In addition, in cases of trauma, oedema, or traction, small and less mobile nerves, such as AIN, have been shown to be at greater risk of injury and avulsions than bigger and more mobile nerves such as the main median nerve trunk [33,34,35].

Clearly differentiated from PS, anterior interosseous nerve syndrome (AINS) has been presented. Duchenne [36] reported one case of isolated palsy of the FPL already in 1872, but Kiloh and Nevin first described AINS in 1952 [37]. It is characterized by weakness or paralysis of FPL, FDP2, and PQ without sensory changes. Many authors discriminate an incomplete AIN palsy from a complete AIN palsy, and the term pseudo-AINS is also used for incomplete cases. It is debated whether AINS is a compression neuropathy or not.

### 2.6. Other Compressive Structures

On rare occasions, patients have been described to have other compression points, including vascular structures, thrombosis of crossing vessels, enlarged bicipital bursa, scar, hypertrophic brachialis muscle, and anomalous muscles surrounding the median nerve [11,15,16,17,20,38,39].

### 2.7. Distal Course of the Median Nerve

The palmar cutaneous branch of the median nerve (PCBMN) emerges about 5 cm proximal to the wrist crease. It usually runs between the tendons of FCR, and PL. PCBMN gives sensory branches to the skin at the palm and thenar area. Anatomical variations exist. The main median nerve then continues inside the carpal tunnel dividing into a motor branch to the thenar and a sensory branch to the thumb, index, long, and radial side of the ring finger.

### 2.8. Other Conflicting Factors

#### 2.8.1. Martin–Gruber Anastomosis

Martin–Gruber anastomosis is an anomalous nerve connection from the median nerve to the ulnar nerve. It is prevalent in about 20% of the population [40]. Multiple branching variations have been described, but the two most common are from the main median nerve trunk before the pronator teres branch and from the AIN branch. Its significance appears in high ulnar nerve injuries when the normal median nerve can preserve a better distal sensory and motor function in the ulnar nerve distribution. Conversely, neuropathy in the median nerve can sometimes lead to symptoms in the ulnar part of the hand. Therefore, proximal median nerve compression can result in more diverse symptoms that would normally be expected [31,41].

#### 2.8.2. Nerve Lesions and Renervation

Classifications for nerve injury have been described by Seddon [42] and Sunderland [43]. However, nerve injury can also be a mixture of the different grades and has a varying potential for recovery. Symptoms of momentary ischemia by external compression resolve quickly, but scarring by prolonged compression or axonal damage requires surgical decompression to enable nerve recovery. More difficult injury takes longer to heal and renervation speed is limited. Chronic denervation leads to permanent muscle fibrosis. Within these physiological boundaries, decisions must be made regarding the need for surgical intervention.

#### 2.8.3. Double Crush Syndrome

Double crush syndrome was first hypothesized in 1973 by Upton and McComas [44]. Their clinical and electrodiagnostic observations led to the suspicion that one compression site of a nerve makes it more susceptible to other compressions. This clinical hypothesis was later confirmed in animal models and humans, as nerve compression leads to changes in axoplasmic flow and decreased transport of neurotrophic substances [45].

Distal median nerve compression, CTS, is more common than PMNC. Some patients with CTS experience pain also in the proximal arm and shoulder. In 1996, Lundborg suggested the term reverse double crush to reflect that a distal compression of a nerve predisposes it to proximal compressions and symptoms [46].

With the understanding of double or multiple crush syndromes, it has also been proposed that because there are many smaller potential sites of compression of the median nerve, all of which are asymptomatic by themselves, the cumulation of these compressions may eventually result in clinical problems for the patient. In addition, more general factors impairing the function of nerves, such as smoking, alcoholism, diabetes, and thyroid disease, are independent crushes themselves that need to be addressed [45].

#### 2.8.4. Neuralgic Amyotrophy

In 1948, Parsonage and Turner reported an unusual syndrome of pain and paralysis around the shoulder in soldiers during the war years [47]. The name neuralgic amyotrophy (NA) was proposed, but named after the physicians, Parsonage–Turner syndrome has also been used for this condition. NA can be presented in multiple ways, but classically sudden pain at the top of the shoulder blade develops, lasting for a few days or weeks, eventually leading to paralysis of the shoulder girdle. Most patients with NA have difficulties in scapulothoracic movements, resulting in scapular winging, typically affecting the long thoracic and suprascapular nerves. Some patients develop median nerve sensory changes, and, like AINS, some patients also develop weakness of the AIN innervated muscles. The pathophysiology of NA remains unknown and likely includes genetic, autoimmune, and mechanical factors [48].

## 3. Clinical Presentation

### 3.1. Symptoms

Differential diagnosis for PMNC should include CTS but also nerve root compression at the cervical spine and compression at the brachial plexus such as TOS. Cervical radiculopathy usually causes more severe pain than PMNC and the pain radiates from the neck to the distal hand. TOS is a condition in which symptoms can consist of nerve, artery or vein compressions. However, due to variations in anatomy and multiple possible sites of compression, the symptoms vary across patients, and diagnosing TOS can be difficult.

The most common symptom in PMNC is pain at the proximal volar forearm, hand or fingers [2,3,4,15,17,19,26,49,50,51], sometimes radiating to the elbow, the axilla, and the head [52]. Patients may complain of aching discomfort, stiffening of the muscles in the forearm, cramping, clumsiness, loss of strength, and tonic flexion position of the fingers [15,17,28,51,52]. Flexion weakness of fingers can be wide-ranging or limited mostly to the AIN-innervated muscles FPL and FDP2 [23,53,54]. Symptoms typically begin insidiously but occasionally rapidly after muscle sprain or an episode of activity [15,17,19]. They might be work-related in various tasks ranging from writing or reading to heavy manual work with forceful forearm rotations and gripping, e.g., when using a screwdriver, carrying heavy objects or hammering [15,26,27,28,50,54]. Such sports as weight training, rowing, and racket games are possible predisposing factors [17,50,55]. In PMNC, symptoms are usually aggravated by activity but can resolve with rest and return when work resumes [28,50]. 

Many patients complain of numbness, tingling or sensory loss in the median nerve distribution area, sometimes in the palmar cutaneous branch of the median nerve [2,3,4,15,17,26,28,49,51,56]. The difference between patients with PMNC and CTS is that with proximal compression patients usually do not experience the nocturnal awakening typically associated with CTS [2,17,26,50,52]. In addition, CTS does not cause paresthesias in the PCBMN. Both in a series of 343 patients with CTS and in another series with 101 patients, about 6% of patients were also diagnosed with PS [2,50]. In a third series, out of 146 patients who had undergone CTR, 13% were later diagnosed with PS [57].

The main symptom described in classic AINS is the partial or full disappearance of pinch grip between the thumb and index finger, usually unilaterally. Loss of PQ strength is more difficult to notice [58]. AIN is a motor nerve, so no sensory alterations occur. Prodromal pain in the shoulder, arm, or elbow may be present. The aetiologies of anatomical compression, physical exertion, repetitive activity, infection, vaccination, pregnancy, supracondylar humerus fracture on children, sleeping on an arm, forearm immobilization, and surgery unrelated to AIN (e.g., shoulder arthroscopy) have been described, but sometimes there is no apparent cause [18,33,35,59,60,61,62,63,64,65].

### 3.2. Clinical Findings

All possible median nerve compression sites should be examined, especially in the carpal tunnel, with tests such as Phalen’s, Durkan’s, or Tetro’s test. Many patients have positive findings for PMNC and CTS simultaneously [17]. Concomitant ulnar nerve compression must be examined [66]. Thenar muscle tenderness can occur because of proximal problems in the median nerve [28]. In rare cases, muscle atrophy in the forearm can appear as well as atrophy of the thenar muscle [15].

All clinical tests for PMNC are subjective reports made by the examiner, which makes the diagnosis difficult and requires clinical experience. Because a troubled nerve has a lower threshold to withstand pressure, clinical tests that compress the nerve even more are designed to provoke patients’ symptoms (Figure 4). Sensitivity and specificity of these tests have not been defined.

A classic finding in AINS is weakness or inability to make an “OK sign” due to dysfunction of FPL and FDP2 (Figure 5) [18,58,59,60]. However, patients with other PMNCs may present with this finding along with other muscle weaknesses [23,28,50]. In addition, AINS is sometimes mistaken for FPL tendon rupture [59].

Patients with difficult aching pain in the forearm may only have minor muscle weaknesses. Careful, thorough muscle testing, compared with the unaffected side, is imperative to get a general perception of the neuropathy (Figure 5, Figure 6, Figure 7, Figure 8 and Figure 9) [23,50,52].

Along with symptom-provoking and muscle-weakness testing, sensory dysfunction must be explored. This can be done with tests for two-point discrimination or Semmes–Weinstein monofilaments from the median nerve innervated fingers and the PCBMN. As with muscle weakness, sensory deficits may be mild.

### 3.3. Electroneuromyography (ENMG)

Special expertise in performing ENMG in PMNC is needed. In PMNC, compression usually causes only occasional ischemia. Therefore, ENMG studies are neither efficient nor reliable in finding PMNC, especially in early symptoms, including pain and paresthesia. Severe clinical findings can only be seen at the late stages of neuropathy [67]. Most studies of PMNC report positive ENMG findings in a minority of patients [1,2,3,4,17,27,51,52,57,68]. However, studies reporting ENMG findings in the majority of patients probably represent more severe cases [15,69].

In patients with clinically diagnosed AIN weakness, ENMG findings are typically clearer but sometimes more wide-ranging than clinically expected, as complete or near-complete denervation can be seen in FPL, FDP2, and PQ [18,58,59,70]. The injury site can be more proximal than initially thought [61,68]. Repeated ENMG studies can track recovery.

### 3.4. Imaging Studies

Some authors suggest radiographs of the distal humerus, possibly revealing the supracondylar process and the possibility of a ligament of Struthers [12,13].

Ultrasound (US) evaluation of the nerve can be used to identify possible compression neuropathy and to assist perineural injections. Changes in nerve caliber and muscle perfusion can help to diagnose median nerve neuropathies if sufficiently severe [1,71,72,73]. 

In determining the aetiology of AINS, magnetic resonance imaging (MRI) might be useful to differentiate between NA and compression and to identify rare tumors in atypical cases [58,70,74,75]. Fascicular (or hourglass) constrictions of unknown origin, potentially trauma, inflammation, or autoimmune, are a common finding in recent imaging and surgical studies of AINS, suggesting neuritis such as NA. Interestingly, in MRI studies these fascicular constrictions can also locate at the upper arm level in median nerve fascicles that distally form the AIN branch [58,70]. Diagnostic imaging might not always be as helpful in decision-making as clinical evaluation in patients with other PMNCs. Of all patients with clinically diagnosed PS, 57% tested positive on US, and only 5% were positive on MRI [57].

## 4. Treatment

Literature on treatment of PMNC consists of heterogeneous hospital-based case series or reports without comparison of treatments. Most of the reports include surgery, but the extent of reporting the state of the nerve and whether thorough decompression of all possible sites of compression was performed varies across the studies. Due to the complexity of PMNCs, the accuracy of diagnoses can understandably be questioned.

### 4.1. Non-Operative Treament

A trial of conservative therapy with activity modifications is advisable. As the focus of management is on relieving pressure over the nerve, forearm pronosupination, gripping, flexing the elbow, and other strenuous repetitive activities should be diminished. Even total cessation of strenuous activity can be tried. Job modifications by changing elbow or forearm position during activity might be useful. Frequent breaks to supinate the forearm are encouraged if the patient works in front of the computer [17,28,50,55,76]. Good results have been reported for up to 70% of patients [28,50].

Injecting local anesthetics [28] or corticosteroids [50] to the PT muscle area has been reported. US-guided injection gave relief of symptoms to PS patients in one series [72].

In patients with AIN weakness, modification of upper and lower arm activity can be made as in other PMNCs. Different supportive treatments could be tried, but most studies report only waiting for spontaneous nerve recovery. In the absence of a clear cause or trauma, most patients with even total AIN palsy can be observed for spontaneous recovery, with good results [18,62]. Still, in up to 30% of patients, weakness or palsy remains [64].

### 4.2. Operative Treatment

Currently, no trials comparing operative and non-operative treatment exist, nor is there a consensus regarding the indications for operative treatment for PMNC. However, the general expert opinion is that surgical release is warranted in PMNC if symptoms are progressive or non-operative treatment is insufficient after a 3-month trial [15,61,62,66,69].

When operating on PMNC, all potential sites of pathology, including the ligament of Struthers, lacertus fibrosus, PT, and FDS arch, should be decompressed [17,51,55]. All fibrotic bands must be released along the course of the median nerve and AIN at the level of proximal forearm and distal arm if needed. PT fascia can be elongated to gain better exposure to the nerve. Afterwards, the compressive site of the nerve presents with the paucity of vasculature distally and increased tortuosity of the vasa nervorum proximally [51].

In a series of 27 patients with PMNC treated by Mackinnon, surgical decompression provided satisfactory outcomes for 93% of patients measured by improvements in strength, pain, quality of life, and DASH scores [51]. In a series of 55 surgically treated patients with PMNC, 80% reported good outcomes and 96% would undergo the operation again. Long progression time of symptoms significantly correlated with persisting symptoms at follow-up. In 51% of patients, the recovery was immediate, and in 49% it took more than 3 months [15]. Good results have been reported, with up to 90% of patients satisfied with the outcome [3,4,6,17,20,26,52,54].

A transverse incision [77] and an endoscopic technique [27,69] have been presented to minimize the length of the scar and to promote faster healing. In the study of Lee and colleagues [69], all 13 endoscopically operated patients received a significant improvement in DASH scores [69]. Still, a limited decompression might increase the risk of residual nerve impingement. With a small transverse incision in the forearm of 21 patients, the outcome for 12 (57%) was excellent or good, for 6 (29%) fair and for 3 (14 %) poor in the series reported by Tsai and colleagues [77]. With a careful clinical examination of the compressive site, many authors use a small incision with wide-awake local anesthesia without a tourniquet (WALANT) to treat LS, allowing perioperative testing of strength recovery after decompression of the nerve [22,23,30].

Patients with one clear mechanical cause or an apparent injury causing the symptoms seem to benefit from surgery the most. In a case series by Seitz and colleagues [19], all 7 patients with an acute injury to the elbow causing a partial rupture of biceps brachii, thus tethering the lacertus fibrosus over the median nerve, showed almost immediate relief of symptoms after surgery [19]. In 44 patients with compression under the lacertus fibrosus, Hagert [23] showed significant improvement after surgery; in most of them the improvement was immediate [23]. In addition, releasing the median nerve under the supracondylar process and the ligament of Struthers led to complete relief of symptoms in most patients compared with patients with more diffuse compression points [8,9,12,13,14].

The literature on surgery for AINS is limited to case reports or series that reveal an inconsistent benefit or varying recovery times. Hill and colleagues reported a series of 24 patients with surgically treated AINS; 8 of these patients returned to full function within 3 months or less, but the rest of them were slower to recover, taking up to 2 years [18]. Ulrich and colleagues noted that 13 out of 14 patients with AINS recovered well after surgery that was performed 12 weeks after initial symptoms [62]. In a case series by Park and colleagues, 11 patients with AINS treated conservatively for at least 6 months with no improvement were then surgically treated; a good outcome was recorded in 10 and a fair outcome in 1 patient [60]. In another series of 15 patients by Schantz, 9 patients improved at 7 weeks postoperatively [59]. Spinner reported full or partial recovery in just 3–12 weeks postoperatively [78]. Interestingly, Hill presented a patient with paralyzed FPL for 4 years who attained full function at 5 weeks after neurolysis [18].

If surgery is needed, a short immobilization with dressings or a splint for a few days after surgery followed by active mobilization is advised. After the wound has healed, a strengthening program is started. Return to work varies but return to sports is advised at 6 to 8 weeks after extensive surgery. Shorter wounds and smaller operations tend to lead to a much faster recovery, often in just a few days [23,27,69].

While adequate surgical management of PMNC usually gives satisfactory results, recurrence of symptoms has been described [17]. Even with multiple sites of compression of the median nerve in the forearm and carpal tunnel, initial management can resolve all the symptoms, at least temporarily. According to the literature on double crush syndrome, releasing one site of compression may revive the nerve for a while even if another remains unreleased, however, recurrence of symptoms is likely [45]. This might be the case with patients with suspected recurrence of CTS [17,79].

## 5. Discussion

CTS is widely diagnosed and treated. As PMNC symptoms can resemble CTS, these two conditions are often confused with each other. As a rarer diagnosis, PMNC might be underdiagnosed and left untreated. An undiagnosed PMNC might explain why all the patients with CTS do not recover after CTR [79]. In addition, due to Martin–Gruber anastomosis, symptoms of PMNC might mimic ulnar neuropathies. This can confuse diagnosing the condition but may also explain residual symptoms after properly executed ulnar nerve release.

CTS and PMNC can coexist in the same patient [2,17]. Olehnik and colleagues described their protocol for treating patients with symptoms of median nerve compression if the diagnosis was not certain for either CTS or PMNC. If ENMG is positive for CTS, they usually recommend only CTR, expecting some of the proximal symptoms to diminish as well. However, if ENMG is negative for CTS, they strongly advise proximal median nerve decompression as the initial procedure [4].

The variability of nerve injury needs to be understood. Compression neuropathies typically cause only mild changes such as neurapraxia or demyelination at most. When clinical findings are minor, PMNC is mostly a clinical diagnosis, and many criticize objective measurements, such as ENMG, to make the diagnosis [23,51]. However, axonopathy can occur in severe cases, which makes ENMG and other imaging studies necessary. In addition, poor recovery postoperatively could be due to residual impingement or more difficult initial nerve injury. The rate of nerve regeneration must be considered when progress is slow.

Whether nerve decompression surgery is beneficial compared with spontaneous recovery has been debated. When repetition and posture are related to onset of PMNC, the treatment should begin conservatively. Work modifications and reduction of strenuous activity can resolve symptoms by lessening muscle compressions over the nerve. Even nerve problems resulting from NA, inflammation, transient compression or trauma could spontaneously recover if the nerve is intact with no scarring. In most studies, if spontaneous recovery failed after a few months, surgery was performed. Due to a lack of randomized trials, the benefit of surgery relative to conservative treatment is unknown.

Treatment of AINS remains controversial due to difficulties in properly defining the cause. In their critical review on AINS, Krishnan and colleagues suggested that AINS is a form of NA and might not originate from compression in the forearm [64]. ENMG findings and imaging nerve fascicular contractions with MRI or US could provide clarification [70,71]. Without any evident compression over the nerve, normal regeneration might take place. With no spontaneous recovery, surgical decompression can be performed, preferably exploring the distal arm, and decompressing all encountered fascicular constrictions [41,64].

However, AINS can also be caused by compression. In a series of 44 patients with LS who benefitted greatly from surgical decompression, Hagert found that the most profound symptom was distinct weakness of muscles FPL, FDP2, and FCR. Only rarely did these patients have distal paresthesia [23]. In reports of AINS, FCR strength is seldom mentioned. AINS might sometimes be labelled too eagerly as neuritis, without a thorough clinical examination to identify a possible site of compression.

Because of potential multiple PMNC sites, some authors suggest separate names for the syndromes to distinguish them. The difference between LS and superficialis/pronator syndrome was discussed by Tang in a recent article suggesting the separation of these two [30]. Both Hagert and Tang encourage a selective mini-invasive technique for decompression with no need to release all potential compression sites. In turn, Sos and colleagues found no isolated compression over the AIN branch, but a compression proximal to it, in patients with classical AINS. They concluded that the proximal median nerve trunk must be explored even in patients without sensory symptoms and stated that there is no interest in distinguishing AINS from PS. Moreover, they criticized specific tests for nerve compression sites because they did not correlate with the actual entrapment sites found in surgery [15]. It is not uncommon to encounter two or more compressive structures over the median nerve in the surgery [3,17]. Mackinnon stated that it is impossible to reliably distinguish the compressive structures preoperatively and recommended a complete surgical release of the median nerve and AIN branch to minimize the risk of recurrence [51].

Tight lacertus fibrosus might cause pain, tightness, and swelling of the proximal flexor-pronator mass during repetitive forearm motion without apparent distal sensation symptoms, referring to an exertional compartment syndrome [80]. On the other hand, recent reviews of chronic exertional compartment syndrome of the forearm described patients characterized by pain in the volar forearm, decreased muscle strength, stiffness, and paresthesia in the fingers [81,82]. ENMG is negative, and patients benefit significantly from fasciotomies. The difference between PMNC and chronic exertional compartment syndrome in the forearm is therefore also questionable. The benefit of surgery could derive from fasciotomy but also from simultaneous nerve decompression.

This review presented available information of symptoms, diagnosis, treatment, and outcome of PMNC. Still, understanding of PMNC is far from complete. Good-quality clinical research is needed. Future research should focus on identifying the natural process and epidemiology of PMNC. The differential diagnosis between compression neuropathy and NA remains unclear. The cause of fascicular constrictions in AIN warrants closer investigations. More detailed information on certain symptoms and related clinical findings would be useful. Treatment guidelines rely on expert opinion and need to be properly addressed. Randomized trials are needed to clarify the need for surgery compared with conservative treatment.

## 6. Conclusions

Due to variability in anatomy and nerve injury, the findings in PMNC can range from subtle weakness to palsy and muscle atrophy. As clear objective signs do not always exist, PMNC is usually a clinical diagnosis with emphasis on tests that provoke symptoms and identify muscle weaknesses. This must be remembered when interpreting ENMG and imaging studies. Uncertainty about the etiology and treatment of AIN-related paresis remains, with some cases apparently caused by compression and others not. Sensory and motor symptoms of PMNC can mimic CTS, and this should be borne in mind, especially if CTR does not resolve all CTS symptoms in a patient.

A nerve injury regardless of etiology can recover spontaneously provided that the nerve remains intact, but pros and cons of surgery should be carefully and individually weighed if no recovery occurs. If needed, surgical decompression of all possible compression sites can yield satisfactory results.

In PMNC, several simultaneous compressive structures might exist that could go unnoticed with an overly limited view. Therefore, the need to separate PMNCs into smaller anatomy-based syndromes can be questioned, as none of them truly represent the mixture of overlapping anatomical variations encountered in clinical practice. Before clear progress in clinical research can be achieved, proximal median nerve-related pain and motor and sensory symptoms of multiple different etiologies should be named more comprehensively as a proximal median nerve syndrome.

## Figures and Tables

**Figure 1 jcm-11-03988-f001:**
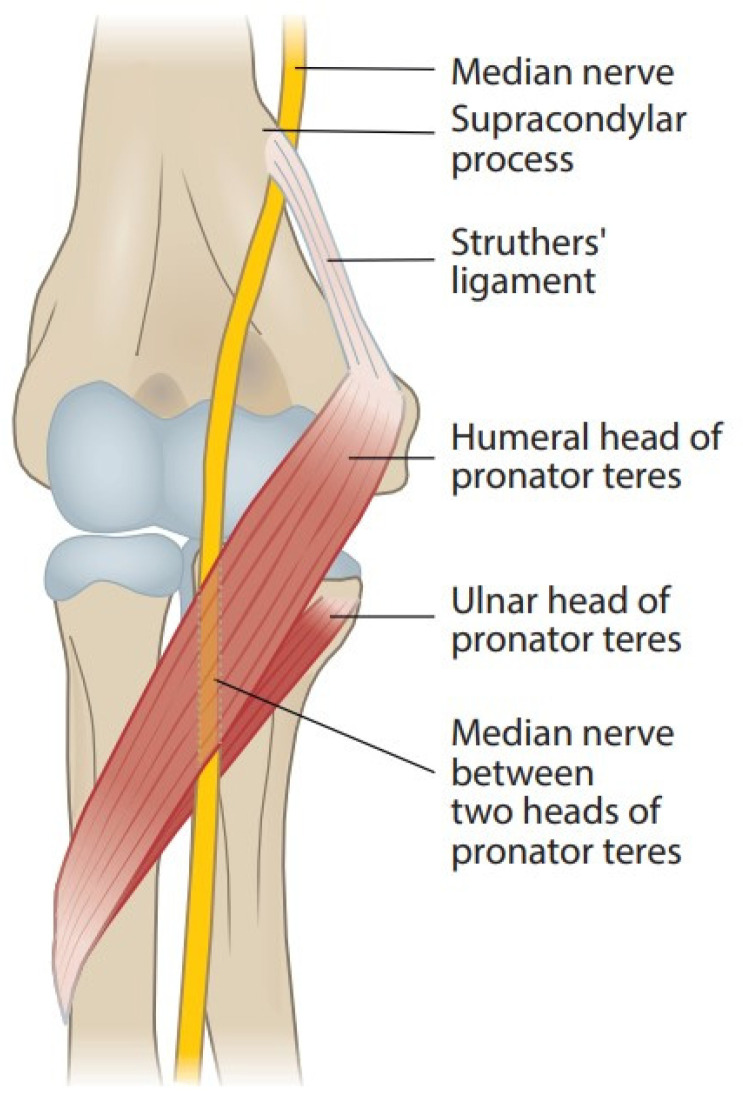
Course of the median nerve at the elbow.

**Figure 2 jcm-11-03988-f002:**
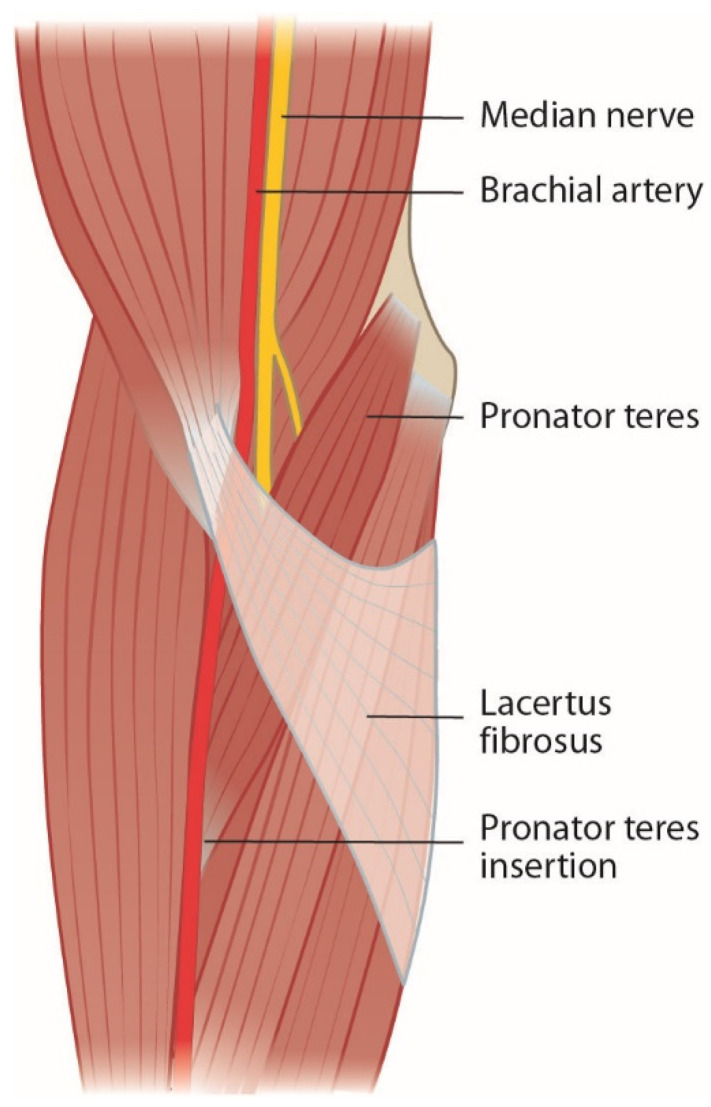
Lacertus fibrosus covers the pronator-flexor mass and the median nerve.

**Figure 3 jcm-11-03988-f003:**
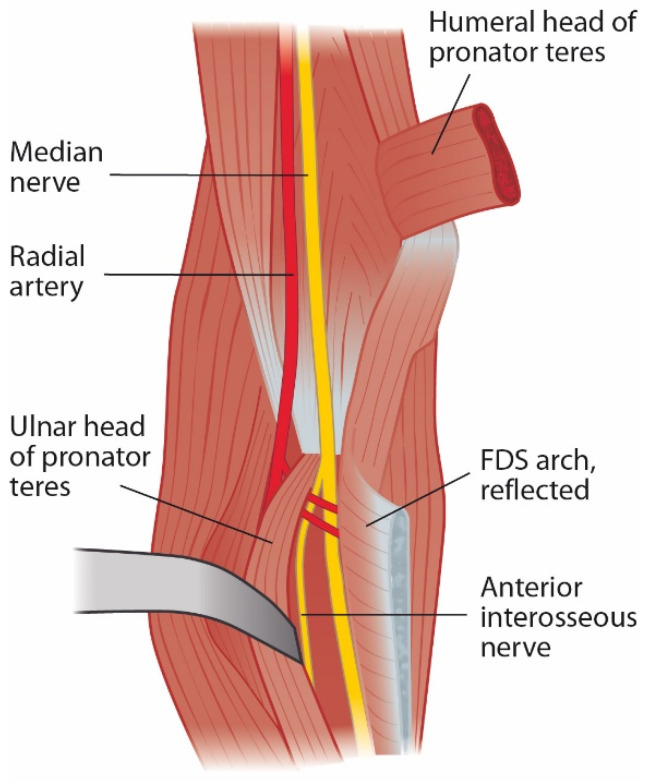
Leading tendinous edge of the flexor digitorum superficialis arch can cause compression of the median nerve and anterior interosseous nerve.

**Figure 4 jcm-11-03988-f004:**
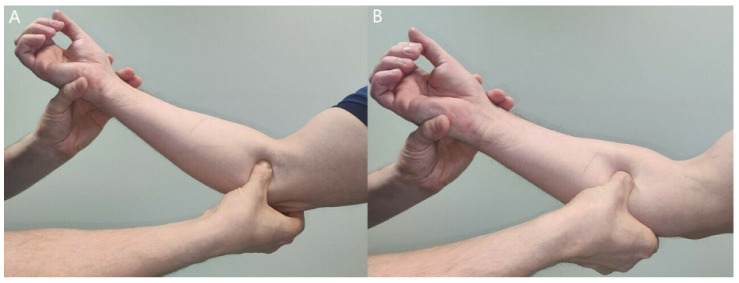
Manual compression of the median nerve at lacertus fibrosus (**A**) or at pronator teres and FDS arch (**B**) produces local pain and even distal paresthesia, indicating a nerve compression at that level. Compression test must also be performed at the Struthers’ ligament proximal to the elbow joint level.

**Figure 5 jcm-11-03988-f005:**
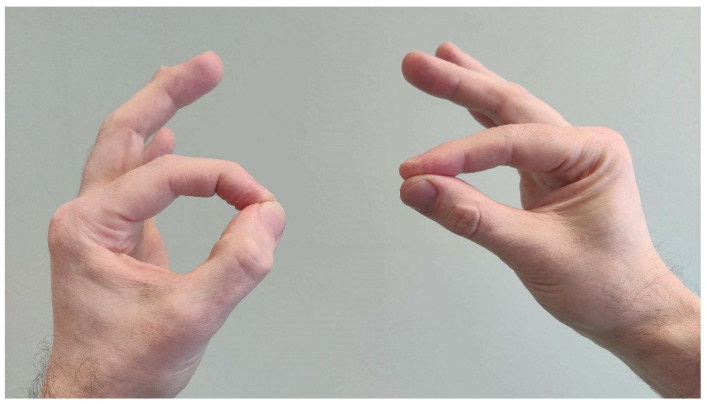
FPL and FDP2 weakness in the right hand and inability to make the OK sign indicates nerve injury proximal to the muscles mentioned.

**Figure 6 jcm-11-03988-f006:**
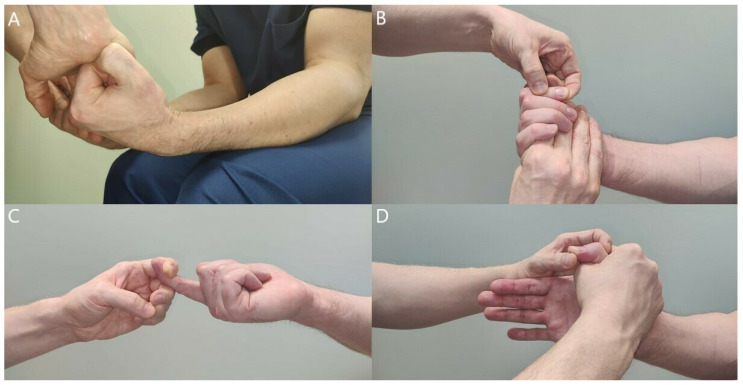
Resisted wrist (**A**), FDP2 (**B**,**C**), and FPL (**D**) flexion reveals minor weaknesses that the patient might not have yet registered. Weakness in the muscle indicates compression of the median nerve proximal to the site of muscle innervation. All tests must be compared with the unaffected side.

**Figure 7 jcm-11-03988-f007:**
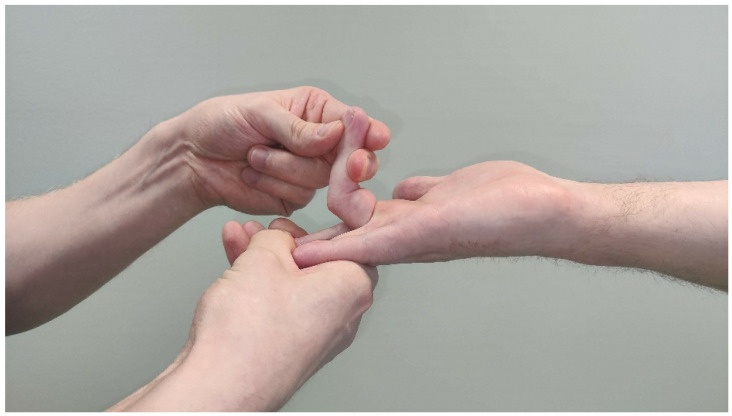
Resisted FDS3 flexion causes pain proximal in the forearm, indicating a nerve compression at the FDS arch.

**Figure 8 jcm-11-03988-f008:**
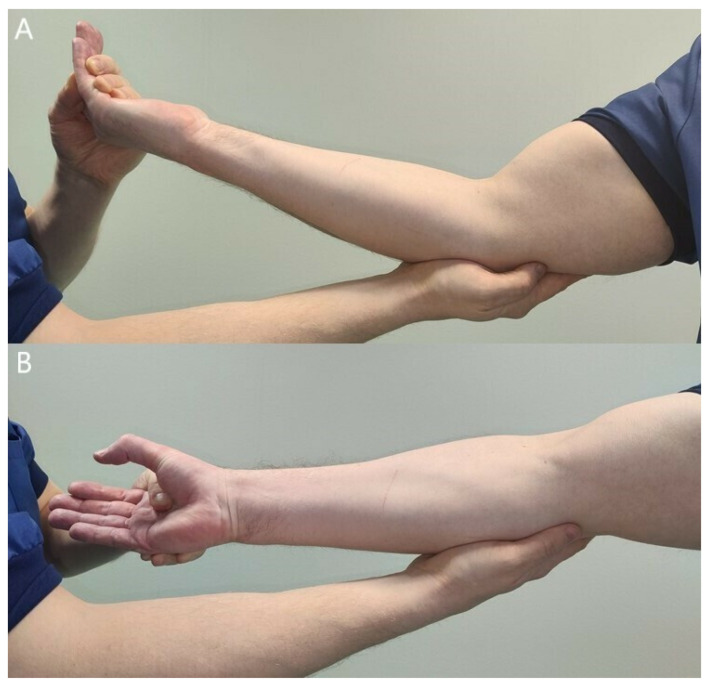
Resisted elbow flexion tightens the lacertus fibrosus and compresses the median nerve, causing local pain and sometimes distal paresthesia (**A**). Resisted forearm pronation in full supination tightens the pronator teres and compresses the median nerve (**B**). Local pain and a loss of pronation power can be observed compared with the unaffected side.

**Figure 9 jcm-11-03988-f009:**
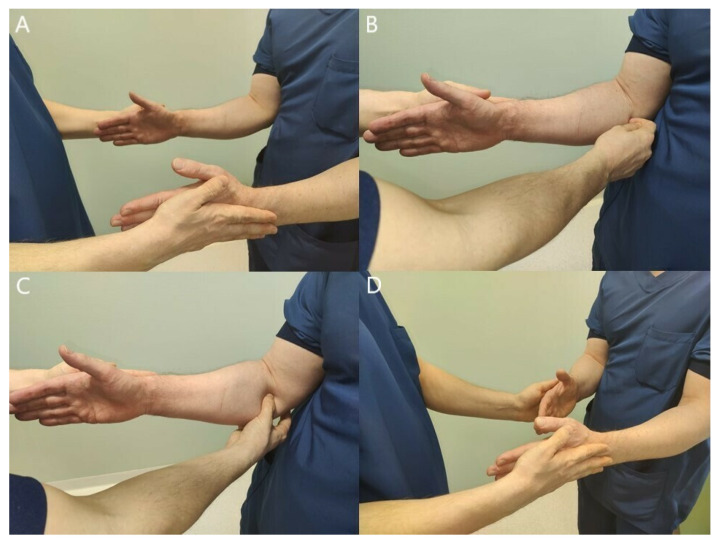
Scratch collapse test. With the patient’s elbows flexed at 90 degrees and the arms held at the sides, the patient externally rotates the arms while the examiner resists the movement (**A**). After releasing the resistance of the rotation, the skin on top of the median nerve is scratched (**B**). Instead of scratching the skin, the examiner can compress the median nerve at the point of maximal tenderness (**C**). After the median nerve irritation, the patient is temporarily unable to resist the rotating force and the affected arm collapses, indicating a proximal compression of the nerve (**D**).

## Data Availability

Not applicable.

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
