# Peer review of "Proximal Median Nerve Compression in the Differential Diagnosis of Carpal Tunnel Syndrome"

_jcm, 2022, doi:10.3390/jcm11143988_

Round 1

Reviewer 1 Report

TItle:

 Perhaps there is another title that would be more appealing and above all concrete of the article.

The review does not only talk about the clinical pathology of the median. You should improve it and say if it is a systematic review. It should indicate that this article talks about neuropathies from the shoulder to the distal area of the upper limb.

Abstract:

Line 18- It should indicate the methodology of the review in the abstract.

Line 19- It should also indicate the main results of the review.

Keywords. Check them as some are not part of the pubmed MeSH thesaurus and should be.

Introduction

Line 41- Comment what type of review it is, not what abstract.

Methodology sub-section

Line 42- You should add a subsection on methodology with a description of the searches carried out with the databases used, the keywords and the exclusion and inclusion criteria of the search as well as the initial and final articles after applying the criteria, also indicate a flowchart of this methodology carried out.

Line 43- You must add in this methodology section whether the PRISMA criteria for sitematic reviews have been followed.

Suitable sections 2, 3 and 4.

Discussion

In general, reorder the concepts in this section.

Line. 408- indicate whether you think you have succeeded in meeting the objectives of your review.

Line 436- maybe it is appropriate to talk about the clinical first, starting on line 409, and the conclusions you draw about them after your review and after the treatment as you start on these lines but at the beginning of the discussion and then to elaborate on the surgery.

Line 474- Talk about whether it meets the quality criteria and whether the studies found are sufficiently rigorous to provide the conclusions you summarise in the article.

Conclusions.

Try to summarise them as much as possible.

Author Response

TItle:

 Perhaps there is another title that would be more appealing and above all concrete of the article.

The review does not only talk about the clinical pathology of the median. You should improve it and say if it is a systematic review. It should indicate that this article talks about neuropathies from the shoulder to the distal area of the upper limb.

-We thank the reviewer for these important comments. The main idea of this article is to present differential diagnoses for carpal tunnel syndrome by introducing proximal median nerve compressions. The title was changed to “Proximal median nerve compression in the differential diagnosis of carpal tunnel syndrome”.

-This is a thorough, but not a systematic review per se. To keep clinical orientation a narrative review was chosen instead.

-Following explanation of the methodology used was added to the text:

A clinically oriented literature search was performed in November 2021 in PubMed/Medline. Terms ``Pronator syndrome”, “Lacertus syndrome”, “Supracondylar process syndrome”, “Proximal median nerve compression” and “Anterior interosseous nerve syndrome” were used. Abstracts of published articles were read, and a full-text version was reviewed if they seemed relevant to the subject. All papers reporting data such as symptoms, clinical diagnosis, treatment, or outcome after treatment were selected. Articles focusing on only diagnostic studies and presenting no clinical features were not included. Only articles published in the English language were included. Only articles accessible at our facility were included.

Abstract:

Line 18- It should indicate the methodology of the review in the abstract.

-This was added to the abstract as “clinically oriented review”.

Line 19- It should also indicate the main results of the review.

-We feel that a better understanding of the proximal problems along with the median nerve help clinicians in differentiating CTS and the proximal compressions. The authors suggest in the concluding chapter that the name “proximal median nerve syndrome” should be used in the future

Keywords. Check them as some are not part of the pubmed MeSH thesaurus and should be.

-These were corrected.

Introduction

Line 41- Comment what type of review it is, not what abstract.

-This was corrected as “narrative review”.

Methodology sub-section

Line 42- You should add a subsection on methodology with a description of the searches carried out with the databases used, the keywords and the exclusion and inclusion criteria of the search as well as the initial and final articles after applying the criteria, also indicate a flowchart of this methodology carried out.

-This information was corrected in the manuscript. As this is not a systematic review, we didn’t add a flow chart. We went through the articles as clinicians and did our best to sum up the relevant information provided by them in a narrative and understandable way.

Line 43- You must add in this methodology section whether the PRISMA criteria for sitematic reviews have been followed.

-As this is a narrative review PRISMA guidelines were not used.

Suitable sections 2, 3 and 4.

Discussion

In general, reorder the concepts in this section.

Line. 408- indicate whether you think you have succeeded in meeting the objectives of your review.

-This information was added.

Line 436- maybe it is appropriate to talk about the clinical first, starting on line 409, and the conclusions you draw about them after your review and after the treatment as you start on these lines but at the beginning of the discussion and then to elaborate on the surgery.

­-We don’t fully understand the meaning of this sentence. Could you give more precise comments?

Line 474- Talk about whether it meets the quality criteria and whether the studies found are sufficiently rigorous to provide the conclusions you summarise in the article.

-Evidence is of low quality but suggests the careful conclusions stated in the manuscript. We added, “This review presented available information of symptoms, diagnosis, treatment, and outcome of PMNC.”

Conclusions.

Try to summarise them as much as possible.

-We have now revised the text.

Reviewer 2 Report

1-The statements regarding the confusion of CTS with PMNC were used frequently inside the manuscript. However, there are several distinct features of CTS and PMNC and clinicans may not necessarly confuse two entities. 

2-Radiculopathy may be confused with PMNC. A brief explanation on the discriminative factors can be added.

3-Are the figures/illustrations original? 

4-Conservative management of the conditions should be given in more detail. 

5-The role of the funder should be explained. 

Author Response

1-The statements regarding the confusion of CTS with PMNC were used frequently inside the manuscript. However, there are several distinct features of CTS and PMNC and clinicans may not necessarly confuse two entities. 

2-Radiculopathy may be confused with PMNC. A brief explanation on the discriminative factors can be added.

3-Are the figures/illustrations original? 

4-Conservative management of the conditions should be given in more detail. 

5-The role of the funder should be explained. 

We thank the reviewer for these important comments.

  1. We brought up the possible confusion between CTS and PMNC to remind the readers of the presence of PMNC in this Journal’s Special Issue of CTS. It is important to remind readers of this Special Issue that PMNC is more commonly confused with CTS than one might think such as in cases of only partial recovery or even symptom recurrence after carpal tunnel release. It is true that most cases are easier to diagnose, and these distinctive features are discussed in the manuscript.
  2. Proximal median nerve neuropathies such as cervical radiculopathy and thoracic outlet syndrome were intentionally left out to shorten the manuscript. A short explanation was added to chapter 3.1:

“Differential diagnosis for PMNC should include CTS but also nerve root compression at the cervical spine and compression at the brachial plexus such as TOS. Cervical radiculopathy usually causes more severe pain than PMNC and the pain radiates from the neck to the distal hand. TOS is a condition in which symptoms can consist of nerve, artery, or vein compressions. However, due to variations in anatomy and multiple possible sites of compression, the symptoms vary across patients, and diagnosing TOS can be difficult.”

  1. Figures are original photographs and drawings.
  2. There is no relevant evidence-based knowledge of conservative treatment of PMNC. Small patient series that were referred to in the manuscript include varying information concerning the treatment. We expanded chapter 4.1 a bit to cover the conservative treatment in more detail.
  3. University of Helsinki, the employer of S.H. and J.R. will fund the journal’s open access and article processing charge (APC). A Finnish Medical Foundation (a non-profit organization) grant was used to enable author S.H. participation in the study on research leave from otherwise full-time clinical work.

Reviewer 3 Report

Dear author,

Thank you for the opportunity to read this interesting work. You have done an excelent job but some improvement are needed. For example is important to discuss more about neural lesion in elbow fracture, especially in children. Here is an article that debates the neurovascular abnormalities in supracondylar fractures in children  Bălănescu, R.; Ulici, A.; RoÈ™ca, D.; Topor, L.; Barbu, M. Neurovascular Abnormalities in Gartland III Supracondylar Fractures in Children. Chirurgia 2013108, 241–244

Author Response

Dear author,

Thank you for the opportunity to read this interesting work. You have done an excelent job but some improvement are needed. For example is important to discuss more about neural lesion in elbow fracture, especially in children. Here is an article that debates the neurovascular abnormalities in supracondylar fractures in children  Bălănescu, R.; Ulici, A.; RoÈ™ca, D.; Topor, L.; Barbu, M. Neurovascular Abnormalities in Gartland III Supracondylar Fractures in Children. Chirurgia 2013108, 241–244

-The main idea of this article is to present differential diagnoses for carpal tunnel syndrome by introducing proximal median nerve compressions. It is true that elbow fractures cause median nerve injuries but expanding the manuscript too far to treating fractures might lose the reader’s focus. Nerve injuries in elbow fractures are usually limited to the AIN branch as is already discussed in chapter 2.5. To emphasize this important aspect in children further, we now changed the word “trauma” to “supracondylar humerus fracture on children” in chapter 3.1.

The article by Bălănescu and colleagues presents a series of 80 patients with supracondylar humerus fractures allocated to two different surgical interventions. The article presents neurovascular complications after this treatment and excludes primary neurovascular complications. It is true that surgery itself can increase the risk of a nerve injury regardless of the site of the surgery, but in this example, the fracture itself must of course be kept as the primary risk. As pointed out earlier, we don’t feel it’s important to go too deeply into this topic in this manuscript. But to shortly emphasize this risk of nerve injury in children due to the fracture these articles were added as a reference:

Vincelet, Y.; Journeau, P.; Popkov, D.; Haumont, T.; Lascombes, P. The Anatomical Basis for Anterior Interosseous Nerve Palsy Secondary to Supracondylar Humerus Fractures in Children. Orthop Traumatol Surg Res 2013, 99, 543–547, doi:10.1016/j.otsr.2013.04.002.

Joist, A.; Joosten, U.; Wetterkamp, D.; Neuber, M.; Probst, A.; Rieger, H. Anterior Interosseous Nerve Compression after Supracondylar Fracture of the Humerus: A Metaanalysis. J Neurosurg 1999, 90, 1053–1056, doi:10.3171/jns.1999.90.6.1053.

We thank the reviewer for this important comment.

Round 2

Reviewer 1 Report

Dear Authors,

The subject of the article is necessary and interesting, but the methodological flaw is important.

A systematic review is needed in order to meet the objectives of clinical differential diagnosis proposed. It could be that if there is no rigour in the chosen studies, the diagnostic manoeuvres could give false positives or negatives and we would not be able to assess to what extent or percentage.

Best wishes

Author Response

Dear Reviewer,

Thank you for your valuable comments.

The authors agree that a systematic review would generally be scientifically the best way to sum up medical literature. However, the proximal median nerve compressions are rare, and the literature consists of case reports and small case series. Actually, at first, we tried to approach this topic in a systematic review manner, but with the poor quality of literature, we would end up with no studies at all. In this difficult clinical problem, our approach provides the most useful information available to the clinicians.

Reviewer 2 Report

Thank you for addressing the queries.

Author Response

Thank you for your comments and suggestions.

Reviewer 3 Report

No further comments.

Author Response

(The authors gave the same response as above.)
